# High-Throughput Effect-Directed Monitoring Platform for Specific Toxicity Quantification of Unknown Waters: Lead-Caused Cell Damage as a Model Using a DNA Hybrid Chain-Reaction-Induced AuNPs@aptamer Self-Assembly Assay

**DOI:** 10.3390/s23156877

**Published:** 2023-08-03

**Authors:** Jiaxuan Xiao, Kuijing Yuan, Yu Tao, Yuhan Wang, Xiaofeng Yang, Jian Cui, Dali Wei, Zhen Zhang

**Affiliations:** 1School of Emergency Management, School of the Environment and Safety Engineering, Jiangsu University, Zhenjiang 212013, China; jiaxuanup@163.com (J.X.); sycyzjf@126.com (Y.T.); yvhanwong@gmail.com (Y.W.); xiaofengsix@163.com (X.Y.); wdl2211609008@163.com (D.W.); 2Dalian Center for Food and Drug Control and Certification, Dalian 116037, China; ralla_1982@163.com; 3Institute of Botany, Jiangsu Province and Chinese Academy Sciences (Nanjing Botanical Garden Mem. Sun Yat-Sen), Nanjing 210014, China; cuijianx@163.com

**Keywords:** biosensor, environmental analysis, signal amplification, rapid detection, aptamer

## Abstract

A high-throughput cell-based monitoring platform was fabricated to rapidly measure the specific toxicity of unknown waters, based on AuNPs@aptamer fluorescence bioassays. The aptamer is employed in the platform for capturing the toxicity indicator, wherein hybrid chain-reaction (HCR)-induced DNA functional gold nanoparticle (AuNPs) self-assembly was carried out for signal amplification, which is essential for sensitively measuring the sub-lethal effects caused by target compounds. Moreover, the excellent stability given by the synthesized DNA nanostructure provides mild conditions for the aptamer thus used to bind the analyte. Herein, ATP was treated as a toxicity indicator and verified using lead-caused cell damage as a model. Under optimized conditions, excellent performance for water sample measurement was observed, yielding satisfactory accuracy (recovery rate: 82.69–114.20%; CV, 2.57%–4.65%) and sensitivity (LOD, 0.26 µM) without sample pretreatment other than filtration, indicating the method’s simplicity, high efficiency, and reliability. Most importantly, this bioassay could be used as a universal platform to encourage its application in the rapid quantification of specific toxicity in varied sources of samples, ranging from drinking water to highly contaminated wastewater.

## 1. Introduction

Chemical target analysis is currently considered an important technique for obtaining contaminant information from aquatic environments [1]; based on this information, risk assessments would be implemented according to the target substances after their structures were identified, the concentrations detected, and toxicological data obtained. However, these approaches only focus on preselected pollutants and always ignore the more unexpected drivers of toxicity [2,3]. As an alternative, some new strategies have been proposed, such as high-resolution mass-spectrometry-based nontarget analysis, which could ascertain the environmental occurrence and risk of pollutants through the holistic screening of unknown waters, which would probably include multiple complex environmental matrices. However, various defects limited its further application, including misidentification, false positives, and an insufficiently detailed pollutant database [2,4,5]. In addition, due to a lack of toxicological data about certain substances, the current technology cannot measure the specific toxicity of unknown waters for an accurate health risk assessment [6]. Therefore, the fabrication of a universal platform to address these problems is challenging but urgently needed.

Fortunately, effect-directed analysis (EDA) opens up huge opportunities for change, owing to its unique character in directly evaluating toxic mixtures. It is mainly composed of three key procedures: fractionation, an effect assay, and chemical analysis [7,8]. This approach could distinguish the drivers of toxicity from a tremendous range of environmental matrices for site-specific risk assessments [3]. Although important, EDA is unable to directly provide more detailed information about the drivers of measured effects [9,10], and its potential cannot be fulfilled within a limited timeframe due to its sophisticated fractionation procedures related to toxicological endpoints [11]. This is crucial in practical contexts, such as during an acute public crisis caused by unknown chemicals.

To overcome the limitations mentioned above, aptamer-based fluorescent bioassays were employed because of their excellent stability and specificity [12], whereby the aptamers that are used can selectively capture targets via hydrogen bonding, electrostatic interactions, van der Waals forces, and stacking interactions [5,13,14,15]. Most importantly, owing to their nucleic acid-based molecular recognition elements, which can be selected from random sequence pools by in vitro selection, aptamers show some advantages over antibodies regarding the development of rapid tests [16,17]. In addition, considering that sub-lethal effects are caused by unknown toxic substances at low environmental concentrations, aptamer sensors that are integrated with DNA-functionalized gold nanoparticles (AuNPs) could improve the sensitivity of these bioassays [12,18,19,20,21]. Moreover, the DNA fluorescent probes immobilized on the AuNPs’ surface can result in excellent performance, including high-affinity target binding and enhanced enzymatic stability [22].

In the present study, a AuNPs@aptamer fluorescent bioassay was fabricated to directly measure specific toxicity in unknown waters. The toxicity level of the test sample was quantitated using a toxicity indicator (with ATP used as a model), which could be recognized by an aptamer, and the measured value could be validated using a known toxic substance (Pb^2+^ in the present study). In addition, a relationship between the linear slope of the toxicity indicator and the Pb^2+^ concentration was established through the sub-lethal effect fitting results, which convert the unknown toxicity into the equivalent concentration of Pb^2+^. Furthermore, as a universal analytical platform, the aptamers could be changed according to the relevant requirements, implying the great potential of this bioassay for the rapid quantification of specific toxicity in unknown environmental samples.

## 2. Materials and Methods

### 2.1. Reagents and Materials

Chloroauric acid (HAuCl_4_·4H_2_O) was obtained from the Shanghai Chemical Reagent Company (Shanghai, China). Trisodium citrate was provided by the Sinopharm Chemical Reagent Co., Ltd. (Shanghai, China). The DNA sequences, ATP, GTP, UTP, CTP, and phosphate-buffered saline (PBS) were purified by Sangon Biotechnology Co., Ltd. (Shanghai, China). All other reagents were of analytical grade, and the aqueous solutions that were used were prepared with ultrapure water (18 MΩ, Milli-Q, Millipore, Shanghai China).

### 2.2. Apparatus

Transmission electron microscopy (TEM) and scanning electron microscopy (SEM) were employed to acquire FT-IR spectral images using a microscope (Model: JEM-2100, JEOL, Tokyo, Japan. Model: S-4800, Hitachi, Tokyo, Japan). In addition, a ZS90 Nano Zetasizer (Malvern, Shanghai, China) was used to analyze the surface charge properties of the related materials. Meanwhile, the UV-vis absorption spectra were recorded using U-V 3900 spectrophotometers (Hitachi, Tokyo, Japan), while the fluorescence spectra were measured with a microplate reader (SynergyH5, Shanghai, China).

### 2.3. Functionalization of AuNPs with DNA

The synthesis of AuNPs-DNA was carried out as previously reported, with minor modifications [23]. ATP aptamer (1.3 µL, 100 µM), blocking strand (1.3 µL, 100 µM), and linker strand (0.4 µL, 100 µM) were mixed with 100 µL of citrate-AuNPs (10 nm and 10 nM). After vigorous shaking, the mixture was placed in a laboratory refrigerator (set at −20 °C) and kept for 2 h, then thawed at room temperature. Through a step of centrifuging (12,000 rpm for 10 min), unbounded DNA was removed by washing three times with 1 × PBS, then the AuNPs-DNA mixture was resuspended in the HCR buffer to make a stock solution, which was used for the subsequent experiments.

### 2.4. Preparation of Gold Nanoparticle-Aptamer Polymers (GAPs)

In detail, hairpin DNA probes (H1 = H2 = 5 µM) were dissolved in HCR buffer and kept for 5 min at 95 °C, then immediately cooled to 4 °C and maintained for 1 h using a thermal cycler. Subsequently, the above solution was incubated with an initiator strand at 37 °C for 90 min (the final concentration of H1, H2, and the initiator was 3 µM:3 µM:5 µM). After the HCR reaction occurred, 100 µL of AuNPs-DNA (10 nM) was introduced into 1000 µL of the above-mentioned reaction mixture and incubated at 37 °C with gentle shaking in the dark for 2 h. The final product (GAPs) was washed three times (6000 rpm for 10 min) with PBS to remove the unreacted material and was finally redispersed in reaction buffer for further use.

### 2.5. Gel Electrophoresis

To verify the HCR system, 1 µM hairpins (H1, H2) were mixed with a 200 nM initiator strand and incubated for 2 h at 37 °C in HCR buffer, then 2% (*w*/*w*) agarose gel containing 5 µL Gel Red was prepared by using a 1× TBE (50 mM of tris-boric acid and 10 mM of EDTA; pH 8.0). After loading a sample containing 10 µL of each reaction sample and 1 µL of 10× loading buffer, the gel was preformed at 110 V for 40 min, visualized under UV light, and finally, photographed. To confirm the HCR-induced GAPs aggregation, AuNPs-DNA, H1 + H2 + AuNPs-DNA, and H1 + H2 + initiator + AuNPs-DNA solutions were loaded onto a 0.5% agarose gel, run at 100 V for 30 min. Since the AuNPs-DNA were colored, their migration could be traced without the need for DNA dyes.

### 2.6. Fluorescence Measurements

First, 10 nM of GAPs was added to 2 mM of ATP solution (the stock solution was 100 mM). After the reaction took place at 37 °C for 0.5 h, the fluorescence of Cy3 was collected between 540 and 700 nm, using the maximal excitation wavelength at 513 nm.

### 2.7. Cell Culture

HeLa cells were cultured in Dulbecco’s modified Eagle’s medium (DMEM), supplemented with 10% fetal bovine serum (FBS), penicillin (100 units mL^−1^), and streptomycin (100 µg mL^−1^), in a humidified atmosphere containing 5 wt %/vol CO_2_ at a temperature of 37 °C.

### 2.8. Sampling Preparation and Treatment

All actual water samples were collected from the central city of Zhenjiang, in Jiangsu Province. For the cell exposure experiments, 50 mL of each sample was sterilized via an autoclave and filtered through a 0.22-µm polyethersulfone (PES) membrane (Millipore, Munich, Germany). The filtered water samples were mixed with 10 × concentrated DMEM medium (1:9) and adjusted to a pH of 7.2−7.4 with a sodium bicarbonate solution (7.5% *w*/*v*), after adding 10% FBS. Subsequently, the sample was stored at −20 °C before use.

## 3. Results and Discussion

### 3.1. Rationale for the Evaluation System

It has been demonstrated that oligonucleotide-modified AuNPs are less susceptible to degradation by nuclease activity than their unmodified oligonucleotide counterparts. This is because the crowding of DNA on a nanoparticle surface results in the steric inhibition of enzyme-binding [24,25]. According to this strategy, an AuNPs@aptamer fluorescence bioassay was designed for direct measurement of the specific toxicity of unknown waters (as displayed in Figure 1), wherein ATP was selected as a cell damage indicator because its level is closely related to that of mitochondria [26]. In addition, the variation in ATP levels caused by Pb^2+^ was used as a known reference toxicant, and a relationship between the linear slope of the ATP and Pb^2+^ concentration was established through the relevant dose-response results, which means that it converts the measured unknown toxicity into the equivalent concentrations of Pb^2+^.

Meanwhile, the programmable oligonucleotide-modified AuNPs derived by HCR have promoted the packing densities of the oligonucleotide and generated a highly aggregated AuNPs shell to enhance the enzyme stability. At the same time, this advanced structure, which was based on the assembled hairpin DNA, remarkably improved the local concentration of aptamer probes, contributing to the better sensitivity of the method. In addition, the quenching efficiency of the superstructure regarding the bound aptamer probes is more intense than that of the single AuNPs, in which their fluorescence can be fully quenched by the surrounding particles on a cooperative basis, greatly diminishing the background signal.

### 3.2. Characterization of GAPs

As illustrated in Figure 1A, for this work, we preformed DNA polymers via HCR and bound the AuNPs@aptamer via complementary DNA hybridization. The successful synthesis of HCR products is a critical factor for controlling the precise and programmable self-assembly of AuNPs into higher-order nanostructures with defined patterns, which was investigated in detail. As shown in Figure 1B, the initiator strand and H1/H2 have been verified using agarose gel electrophoresis. The bright ladder-like bands were observed in the presence of initiators with different concentrations (lanes 4, 5, and 6), indicating that the long-nicked HCR DNA polymers had formed. In contrast, H1 and H2 maintained their stability without an initiator (lane 3). Remarkably, the ladder-like bands in lanes 4–6 demonstrated that many different-length dsDNA products were generated by the HCR reaction, which is consistent with previously reported studies [27,28]. To further the precise control of the synthesis of GAPs, three classes of HCR products with different lengths were carried out by varying the initiator concentration for the polymerization reaction. In comparison with lane 6 (initiator:hairpin = 1:8), lanes 4 and 5 were prepared with higher initiator concentrations (initiator:hairpin = 1:1 and 1:4), which exhibited accelerated mobility in agarose gel. These results demonstrated that the concentration of initiator in the preparation of GAPs needs to be strictly controlled, to ensure the length of the HCR. At the same time, the atomic force microscopy (AFM) results confirmed that H1 and H2 probes could be self-assembled after the addition of initiators, and finally used to obtain a natural liner DNA polymer (Appendix A). The depth of the linear DNA was ~2.7 nm (Appendix A), which matched well with the results recorded in previous works [29].

Meanwhile, confirmatory experiments were implemented to estimate the feasibility of HCR-induced AuNPs aggregates. The TEM image illustrated that the spherical AuNPs were monodispersed in the aqueous phase, with a diameter of approximately 13 nm (Appendix A), which showed no noticeable change in morphology and monodispersal with AuNP probes (Appendix A). The image was obtained after two thiol-terminated single-strand DNA (linker strands and ATP aptamer) and Cy3-labeled reporter DNA (complement with ATP aptamer) were modified on the AuNPs via Au-S bond. However, through incubation with HCR products, the apparent aggregation of AuNP probes was observed (Figure 1C) when integrated with the gel image of the electrophoretic analysis results (Figure 1D), implying the success of the assembly of GAPs with the long dsDNA.

The synthesized GAPs were further verified via UV-absorbance and DLS. The characteristic absorbance band of the AuNPs appeared from 520 to 529 nm, and its hydrodynamic diameter changed from 22.32 to 321.67 nm (Figure 2F). As shown in Figure 1G, the citrate-coated AuNPs were −14.2 ± 1.2 mV, which increased to −27.3 ± 1.5 mV after DNA modification. When the HCR products were introduced, the potential changed to −23.4 ± 1.8 mV. This was due to the more negatively charged phosphate backbone of DNA, which reduced the extent of zeta potential after the displacement of the citrate with DNA [30]. In addition, further hybridization with HCR products resulted in increased zeta potential, owing to lower-density DNA packing on the outer surface and a rigid double-stranded conformation, which, when integrated with the above analysis results, indicated the success of the assembly of GAPs.

Compared with previous works regarding AuNP aggregation methods using complementary DNA hybridization [31], our proposed approach provides a controllable assembly process with less dense packing, by controlling the linker strand-to-aptamer ratio and the initiator-to-hairpin ratio, respectively. In summary, the above-mentioned results illustrate that the HCR could control the precise and programmable self-assembly of AuNPs into higher-order nanostructures with defined patterns, which confirms the feasibility of the designed method.

### 3.3. The Performance of GAPs Sensor

The accuracy of quantification in the developed analytical system primarily depends on the fluorescence response intensity, stability, and selectivity of the GAPs sensor. Thus, to evaluate the performance of the proposed method, two detection systems were constructed, including the ATP aptamer/blocking strand-Cy3 (hybrid DNA) and AuNPs@ATP aptamer@blocking strand-Cy3 (nanoflare).

Fluorescence response intensity refers to the ability of the analytical system to accurately detect and measure the fluorescence signal generated by the target analyte. This is an essential aspect of quantification as it directly impacts the sensitivity and precision of the system. As illustrated in Figure 2A–C, the results show that all three systems that were exhibited enhanced the fluorescence recovery in the presence of ATP (2 mM). However, the GAPs displayed the most robust recovery intensity, with an F/F_0_ of 12.03, which is better than that of nanoflare (4.12) and hybrid DNA (6.87). These results highlight the fact that the designed ATP aptamer/blocker operates with improved turn-on and turn-off dynamics. Additionally, the high response rate of the GAPs may be attributed to the synergistic quenching effect of the AuNPs and the fluorescence emission of Cy3 being mainly absorbed by themselves and by the neighboring AuNPs. Thus, the present study considers the GAP system to be the optimal design.

In addition, the stability of the proposed GAPs sensor was evaluated by tracking the change in fluorescence intensity following the addition of 20% FBS. Figure 2D shows that the fluorescence intensity gradually decreased for the hybrid DNA and nanoflare, while the GAPs showed minimal fluorescence loss. The above results indicate that the GAPs exhibited excellent resistance against enzymatic degradation in the presence of FBS, owing to the aggregated structure of AuNPs and the dense aptamer modification, resulting in a high salt concentration around the sensor.

The specificity of the GAPs was also assessed by examining the selectivity of ATP compared to its analogs, UTP, CTP, and GTP, at a concentration of 2 µM. As depicted in Figure 2E, the fluorescence recovery of the GAP system was not impacted by the presence of these analogs.

**Figure 2 sensors-23-06877-f002:**
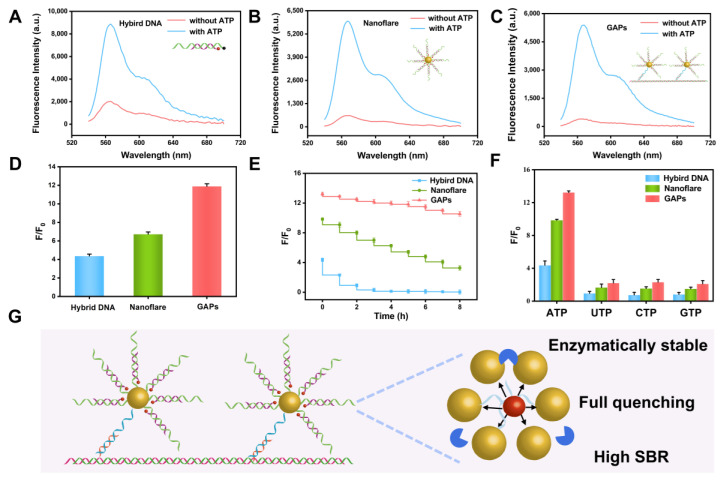
(**A**–**C**) show the fluorescence spectra response of the hybrid DNA, AuNPs nanoflare, and GAPs, respectively; (**D**–**F**) show the fluorescence response intensity, stability, and selectivity of the hybrid DNA, AuNP nanoflare, and GAPs, respectively. (**G**) Schematic illustration of the advantages of GAPs.

In summary, these results highlight the fact that the proposed ATP sensor demonstrates a high signal-to-noise ratio, stability in the fact of enzymes, and specificity, making the sensor a potential tool for detecting ATP in complex environmental matrices.

### 3.4. Optimization of the Method

Several parameters that may affect the method’s performance were evaluated, including the concentration of the linker strand, the initiator-to-hairpin ratio, the reaction time of HCR, the reaction time between the linker-AuNPs and HCR products, and the reaction time between GAPs and ATP.

The initiator strand-to-hairpin ratio determines the fragment length of HCR, which affects the number of bound AuNPs. This is because HCR products that are too long will hinder the binding of linkers onto Au due to crowding, while HCR products that are too short will not be able to achieve the assembly effect. As depicted in Appendix A, the maximum fluorescence intensity was achieved when the initiator-to-hairpin ratio was 1:8. The most appropriate reaction time for HCR was determined to be 90 min, as illustrated in Appendix A. Furthermore, it is very important to optimize the concentration of the linker as it may affect the response of GAPs through steric hindrance. We consider that too much linker will assemble the gold nanoparticles at the center, rather than assembling the gold nanoparticles with the HCR product; however, too little linker does not achieve the effect of assembling the gold nanoparticles. As shown in Appendix A, a systematic increase in the signal response was observed with the increasing concentration of linker DNA, with the optimal concentration being determined to be 1.5 µM. Additionally, a combined duration of 90 min between AuNPs@aptamer@linker DNA and HCR products was optimal, as depicted in Appendix A. The incubation time was also optimized for 30 min, as shown in Appendix A.

Under the optimized conditions given above, the standard curve (Figure 3C) was established after measuring a serial concentration of ATP (Figure 3B). Good performance was obtained with a low limit of detection (LOD, 0.26 µM, and 3 σ/k, where σ denotes the standard deviation obtained by conducting blank samples, and where k is the slope of the calibration curve) and a wide range (3.98 µM–2.6 mM), surpassing the performance of other previously reported methods (Appendix A).

### 3.5. Method Verification and Real Sample Analysis

The accuracy and precision of the GAPs sensor were estimated by measuring water samples spiked with various ATP concentrations. As seen in Table 1, the recovery ranges of the ATP were within 82.69–114.20%, with the intra-assay coefficient of variation (CV) ranging from 2.57% to 4.65%, meaning that the method showed good accuracy and could be used for quantifying the toxicity of unknown water samples. Subsequently, we employed the fabricated sensor to quantify the ATP levels in the cell extracts (HeLa). Figure 3D presents the results of determining the different numbers of HeLa cells. As the concentration of fresh HeLa cell extract improved, the ATP concentration increased.

These results indicate that the developed method could monitor ATP concentrations in cell extracts without interfering with other substances in the cells.

As shown in Figure 3A, to quantify the toxicity of water samples, various concentrations of the reference toxicant Pb^2+^ were introduced into the HeLa cell to incubate, leading to the intracellular ATP level changing with the Pb^2+^ concentration (Figure 3E) and a relationship between the linear slope of the ATP and the Pb^2+^ concentration was established through the sub-lethal effect fitting results (Figure 3F). Based on the fitting results presented above, this method was further applied to investigate potential toxic occurrences in the main rivers of Zhenjiang City, Jiangsu Province, along the lower reaches of the Yangtze River in southeast China, with a population of nearly three million. As shown in Table 2, the results for the equivalent concentration of the reference toxicant Pb^2+^ can be used to assess the overall toxicity of water samples rapidly and efficiently.

## 4. Conclusions

In summary, an AuNPs@aptamer fluorescence bioassay method was fabricated to reliably and efficiently detect toxicity in water based on the HCR-induced aptamer functionalized AuNPs assemble. The established method showed good accuracy and reproducibility (recovery rate: 82.69–114.20%; CV, 2.57–4.65%) with a low LOD of 0.26 µM. These results also support the finding that our method can be used as a universal monitoring platform. These results demonstrate this methodology’s potential utility for assessing the toxicity of unknown water samples. Meanwhile, the present study also provides a novel strategy for designing aptamer-based bioassays for the specific toxicity quantification of unknown waters by other biomarkers with different aptamers.

## Data Availability

Data will be made available on request.

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
