# Peer review of "High-Throughput Effect-Directed Monitoring Platform for Specific Toxicity Quantification of Unknown Waters: Lead-Caused Cell Damage as a Model Using a DNA Hybrid Chain-Reaction-Induced AuNPs@aptamer Self-Assembly Assay"

_sensors, 2023, doi:10.3390/s23156877_

Round 1

Reviewer 1 Report

The paper describes an AuNPs@aptamer fluorescence bioassay method to detect toxicity in water based on the HCR-induced aptamer functionalized AuNPs assemble. It has technical merit. I have the following comments: 

Page 1 line 39 - which defects? 

Page 2 line 64 - elaborate on why AuNPS are used. 

Page 3 line 127 - They need to provide more details about the nature of the samples for example, a waste water treatment plant, sewer,lake, river, tap water etc. Or provide the coordinates of the exact spot where the sample was taken if it was taken from the environment. This observation affect the quality of the whole manuscript since it is not clear where the samples came from. 

Page 4 line 141 - Provide a reference for the statement “Wherein ATP was selected as a cell damage indicator because its level is closely related to mitochondria”. 

Figure 1 - It is very difficult to read the plots E, F, and G. Also, D and C are very small and difficult to see in detail. 

Page 5 lines 203 and 204 - Is there any reference that supports the affirmation “When HCR products were introduced, the potential was changed to -23.4±1.8 mV, which was due to the more negatively charged phosphate backbone of DNA reduced the extent of zeta potential after the displacement of citrate with DNA.”? 

Figure 2 - It is very difficult to read the plots due to the small size. 

Figure 3 - Same as Figure 1 and 2. It is very difficult to read. 

Page 8 line 282 and table 1- How can the recovery be superior to 100%?

Author Response

Review #1

Comments:

The paper describes an AuNPs@aptamer fluorescence bioassay method to detect toxicity in water based on the HCR-induced aptamer functionalized AuNPs assemble. It has technical merit. I have the following comments:

  1. Page 1 line 39 - which defects?

-Answer: Thank you for this comment. At the current stage, high sensitivity low-resolution tandem mass spectrometry is considered as a popular strategy for nontarget analysis in place of traditional techniques, albeit important, it suffers from several defects including (i) expensive instrumentation needed, large volume of each sample to be enriched and complicated pretreatment procedures; (ii) misidentification and false positives (Journal of Exposure Science & Environmental Epidemiology, 2018, 28, 411–426); (iii) insufficient pollutant database(Environ. Sci. Technol. 2021, 55, 3070−3080).

  1. Page 2 line 64 - elaborate on why AuNPs are used.

-Answer: Thank you very much for this thoughtful comment. AuNPs was employed as a key element at this study due to the following reasons: 1) AuNPs are the efficient quenchers against Cy3 by localized surface plasmon resonance (NEST) absorption (Trends Anal. Chem., 2020, 123, 115748); 2) Alkylthiol-functionalized DNA strand can be densely modified on the surface of AuNPs through simple freezing method, enabling to load enough functional strands (J. Am. Chem. Soc. 2017, 139, 9471−9474); 3) AuNPs were used as the cores of the polymers (GAPs) that were encapsulated by oligonucleotide shells, which will improve the stability of the proposed bioassay against the various matrix in real sample measurements (Anal. Chem. 2020, 92, 2379−2382).

  1. Page 3 line 127 - They need to provide more details about the nature of the samples for example, a waste water treatment plant, sewer, lake, river, tap water etc. Or provide the coordinates of the exact spot where the sample was taken if it was taken from the environment. This observation affect the quality of the whole manuscript since it is not clear where the samples came from.

-Answer: Thank you for pointing out this point. According to your good suggestion, the specific spot of natural sampling points was provided and shown in Table S2.

  1. Page 4 line 141 - Provide a reference for the statement “Wherein ATP was selected as a cell damage indicator because its level is closely related to mitochondria”.

-Answer: Done.

  1. Figure 1 - It is very difficult to read the plots E, F, and G. Also, D and C are very small and difficult to see in detail.

-Answer: Figure 1 was modified based on your valuable suggestions.

  1. Page 5 lines 203 and 204 - Is there any reference that supports the affirmation “When HCR products were introduced, the potential was changed to -23.4±1.8 mV, which was due to the more negatively charged phosphate backbone of DNA reduced the extent of zeta potential after the displacement of citrate with DNA.”?

-Answer: Yes, there are some literatures to support this affirmation, which were cited in the revised manuscript.

  1. Figure 2 - It is very difficult to read the plots due to the small size.

-Answer: Sorry for this confusion due to our carelessness, and now the plots were adjusted.

  1. Figure 3 - Same as Figure 1 and 2. It is very difficult to read. 

-Answer: As your kind suggestion, the parts you mentioned were revised.

  1. Page 8 line 282 and table 1- How can the recovery be superior to 100%?

-Answer: Thanks a lot for your good comments, the recovery rate of more than 100% may be caused by the errors of the experimental process, including systematic and human errors (ACS Appl. Mater. Interfaces 2022, 14, 40624–40632; Biosens. Bioelectron., 2012, 34, 106; Spectrochim. Acta. A Mol. Biomol. Spectrosc., 2022, 273, 121044), and its range of 80% to 120% belongs to the normal range (GB/T5750.5-2006).

Reviewer 2 Report

The manuscript titled "High throughput effect- directed monitoring platform for specific toxicity quantification of unknown waters: lead caused cell damage as a model using DNA Hybrid chain reaction induced AuNPs@aptamer self-assembly assay" proposes a high-throughput cell-based monitoring platform was fabricated to accurately measure the specific toxicity of unknown waters based on the AuNPs@aptamer fluorescence bioassay. The paper presents interesting information. However, before being able to recommend this document for publication, some points need to be revised.

The introduction provides a good overview of the current challenges in chemical target analysis and the need for a universal platform to address the limitations of existing approaches. However, it would be helpful to provide a more specific and concise statement of the research problem or gap that your study aims to address. This would help to focus the reader's attention on the main objective of your research.

The introduction introduces the concept of aptamer-based fluorescent bioassays for their stability and specificity. Consider providing a brief explanation of how aptamers work and why they are suitable for the proposed application. This would help the reader understand the rationale behind the choice of this technology.

The last paragraph of the introduction provides a good summary of the objectives and significance of the study. However, it would be helpful to provide a clearer statement of the research hypothesis or research questions that your study aims to address. This would help the reader understand the specific goals of your research.

In the subsection "Functionalization of AuNPs with DNA," it would be beneficial to provide more details about the specific sequences of the ATP aptamer, blocking strand, and linker strand. Including these sequences would enhance the reproducibility of the study.

In the subsection "Gel Electrophoresis," consider providing more details about the gel concentration, running conditions (voltage and time), and visualization method used to detect the HCR-induced GAPs aggregation. These details are important for replicating the experiment and interpreting the results accurately.

In the subsection "Fluorescence Measurements," it would be beneficial to mention the concentration or volume of the ATP stock solution used in the experiment. This information is crucial for understanding the experimental setup and interpreting the fluorescence measurements accurately.

In the subsection "Rationale for the Evaluation System," it would be helpful to provide a brief explanation of the mechanism behind the higher resistance degradation ability of oligonucleotide-modified AuNPs compared to unmodified counterparts. This would give the reader a better understanding of the rationale behind the design of the AuNPs@aptamer fluorescence bioassay.

In the subsection "Optimization of the Method," while the optimized conditions are briefly mentioned, it would be beneficial to provide more details on how each parameter was optimized and the rationale behind the chosen conditions.

Author Response

Review #2

Comments:

The manuscript titled "High throughput effect- directed monitoring platform for specific toxicity quantification of unknown waters: lead caused cell damage as a model using DNA Hybrid chain reaction induced AuNPs@aptamer self-assembly assay" proposes a high-throughput cell-based monitoring platform was fabricated to accurately measure the specific toxicity of unknown waters based on the AuNPs@aptamer fluorescence bioassay. The paper presents interesting information. However, before being able to recommend this document for publication, some points need to be revised.

  1. The introduction provides a good overview of the current challenges in chemical target analysis and the need for a universal platform to address the limitations of existing approaches. However, it would be helpful to provide a more specific and concise statement of the research problem or gap that your study aims to address. This would help to focus the reader's attention on the main objective of your research.

-Answer: Thank you very much for your constructive comments, and the description in the parts (Line 39- 44) provided the information related to the current research problem that we aim to address.

  1. The introduction introduces the concept of aptamer-based fluorescent bioassays for their stability and specificity. Consider providing a brief explanation of how aptamers work and why they are suitable for the proposed application. This would help the reader understand the rationale behind the choice of this technology.

-Answer: Thanks a lot for your suggestions, and the explanation regarding to aptamers recognition mechanism and advantages were added into introduction section (Line 55-60).

  1. The last paragraph of the introduction provides a good summary of the objectives and significance of the study. However, it would be helpful to provide a clearer statement of the research hypothesis or research questions that your study aims to address. This would help the reader understand the specific goals of your research.

-Answer: Thank you very much for this insightful suggestion, and this part has been rewritten to improve the manuscript.

  1. In the subsection "Functionalization of AuNPs with DNA," it would be beneficial to provide more details about the specific sequences of the ATP aptamer, blocking strand, and linker strand. Including these sequences would enhance the reproducibility of the study.

-Answer:  Thanks a lot for this comment, Additional detailed contents of DNA strands are included in the supplementary information (displayed at Table S1).

  1. In the subsection "Gel Electrophoresis," consider providing more details about the gel concentration, running conditions (voltage and time), and visualization method used to detect the HCR-induced GAPs aggregation. These details are important for replicating the experiment and interpreting the results accurately.

-Answer: We thank the reviewer for pointing out this important detail. These data are now provided in the revised version of the manuscript (Line 113-118).

  1. In the subsection "Fluorescence Measurements," it would be beneficial to mention the concentration or volume of the ATP stock solution used in the experiment. This information is crucial for understanding the experimental setup and interpreting the fluorescence measurements accurately.

-Answer: Thank you for this valuable suggestion, and the experimental parameters have been involved in the experimental part (Line 120-121).

  1. In the subsection "Rationale for the Evaluation System," it would be helpful to provide a brief explanation of the mechanism behind the higher resistance degradation ability of oligonucleotide-modified AuNPs compared to unmodified counterparts. This would give the reader a better understanding of the rationale behind the design of the AuNPs@aptamer fluorescence bioassay.

-Answer: We agree with you about this aspect, and the mechanism has been added to this section (Line 138-141).

  1. In the subsection "Optimization of the Method," while the optimized conditions are briefly mentioned, it would be beneficial to provide more details on how each parameter was optimized and the rationale behind the chosen conditions.

-Answer: Thank you for this kind suggestion, and the conditions that you mentioned were supplemented, which could be seen in Line 260-270.
